# The Pre-Operative GRADE Score Is Associated with 5-Year Survival among Older Patients with Cancer Undergoing Surgery

**DOI:** 10.3390/cancers14010117

**Published:** 2021-12-27

**Authors:** Philippe Wind, Zoe ap Thomas, Marie Laurent, Thomas Aparicio, Matthieu Siebert, Etienne Audureau, Elena Paillaud, Guilhem Bousquet, Frédéric Pamoukdjian

**Affiliations:** 1Service de Chirurgie Digestive et Cancérologique, Hôpital Avicenne, APHP, 93000 Bobigny, France; philippe.wind@aphp.fr (P.W.); matthieu.siebert@aphp.fr (M.S.); 2Service d’oncologie Médicale, Hôpital Avicenne, APHP, 93000 Bobigny, France; zoe.ap-thomas@aphp.fr (Z.a.T.); guilhem.bousquet@aphp.fr (G.B.); 3Clinical Epidemiology and Ageing, Université Paris-Est Créteil, IMRB-UPEC/Inserm U955, 94000 Creteil, France; marie.laurent@aphp.fr (M.L.); etienne.audureau@aphp.fr (E.A.); elena.paillaud@aphp.fr (E.P.); 4Département de Médecine Interne et Gériatrie, APHP, Hôpital Henri Mondor, 94000 Creteil, France; 5Service de Gastroentérologie, Hôpital Avicenne, APHP, 93000 Bobigny, France; thomas.aparicio@aphp.fr; 6Service de Santé Publique, Hôpital Henri Mondor, APHP, 94000 Creteil, France; 7Service de Gériatrie, Hôpital Europeen Georges Pompidou, APHP, Paris Cancer Institute CARPEM, 75015 Paris, France; 8Cardiovascular Markers in Stressed Conditions, Université Sorbonne Paris Nord, Inserm UMR_S942, MASCOT, 93000 Bobigny, France; 9Service de Médecine Gériatrique, Hôpital Avicenne, APHP, 93000 Bobigny, France

**Keywords:** cancer, surgery, older adults, postoperative mortality, post-operative complications, prognostic score

## Abstract

**Simple Summary:**

The benefit of major cancer surgery among older patients may be limited, and it remains unclear how to optimally select suitable patients. By combining very simple geriatric (gait speed, and weight loss) and cancer parameters (cancer site and cancer extension), the pre-operative GRADE score > 8 was significantly associated with severe post-operative complications, and overall post-operative mortality among older patients with digestive or non-breast gynaecological cancer undergoing surgery. At the time of the first consultation, the GRADE score could help surgeons to choose the most suitable treatment strategy, avoiding under- or over-treatment, especially when a geriatric assessment is not available.

**Abstract:**

We aimed to assess the prognostic value of the pre-operative GRADE score for long-term survival among older adults undergoing major surgery for digestive or non-breast gynaecological cancers. Between 2013 and 2019, 136 consecutive older adults with cancer were prospectively recruited from the PF-EC cohort study before major cancer surgery and underwent a geriatric assessment. The GRADE score includes weight loss, gait speed at the threshold of 0.8 m/s, cancer site and cancer extension. The primary outcome was post-operative 5-year mortality. Patients were classified as low risk (GRADE ≤ 8) or high risk (GRADE > 8) on the basis of the median score. A Cox multivariate proportional hazards regression model was performed to assess the association between pre-operative factors and 5-year mortality expressed by adjusted hazard ratio (aHR) and 95% CI. The median age was 80 years, 52% were men, 73% had colorectal cancer. The 30-day post-operative severe complication rate (Clavien-Dindo ≥ 3) was 37%. The 5-year post-operative mortality rate was 34.5%. A GRADE score ≥ 8 (aHR = 2.64 [1.34–5.21], *p* = 0.0002) was associated with post-operative mortality after adjustment for Body Mass Index < 21 kg/m^2^ and Instrumental Activities of Daily Living <3/4. By combining very simple geriatric and cancer parameters, the pre-operative GRADE score provides a discriminant prognosis and could help to choose the most suitable treatment strategy for older cancer patients, avoiding under or over-treatment.

## 1. Introduction

With the aging of populations worldwide, the number of older people with cancer is increasing [1], challenging daily surgical practice. Indeed, two-thirds of patients with cancer requiring surgery are over 70 years [2] and these patients are at high risk for under or over-treatment [3]. Data from evidence-based-medicine is lacking for therapeutic decisions because this population is not offered participation in clinical trials [4] or is more likely to be excluded because of limiting comorbidities and dependency [5]. Despite improvements in the surgical management of cancers, 30-day post-operative complications are still highly prevalent among older patients (between 35 and 50%), with mortality rates reaching 13% [6,7], and 1-year mortality reaching 40% [8]. The benefit of cancer surgery among older patients could thus be limited, and it remains unclear how to optimally select suitable patients [9]. 

The main challenge is to select the optimal treatment tailored to patient heterogeneity in terms of social environment, levels of dependency, comorbidities, nutrition, mobility, cognitive and mood status, all liable to lead to post-operative complications and shorter survival [10]. The Geriatric Assessment (GA), which is a multidimensional and multidisciplinary health assessment for older adults [11], is recommended by the International Society of Geriatric Oncology and the European Society of Surgical Oncology before cancer surgery [2,12], but to date there has been no validated decision-based algorithm to determine which older patients could benefit the most from major cancer surgery. For example, the Surgical Risk Calculator (SRC) (http://riskcalculator.facs.org/RiskCalculator/) (accessed on 7 November 2021) which combines 20 items, was applied to 69 older patients with a mean age of 75 years undergoing emergency surgical interventions for a malignant colonic obstruction [13]. In this study, the SRC was an independent risk factor for 30-day post-operative mortality (Odd Ratio = 1.07 [1.01–1.15], *p* = 0.03). More recently, we developed and validated a simple and easy prognostic score (the GRADE score: https://grade.shinyapps.io/dynnomapp/) (accessed on 7 November 2021) to assist therapeutic decisions for older patients with cancer [14]. This score takes into account two geriatric parameters (weight loss and gait speed) and two cancer parameters (cancer site and extension), and predicts short-term mortality (i.e., ≤6 months). We hypothesized that the GRADE score could help decide whether or not to undertake radical cancer surgery in older patients. 

We thus aimed to assess the prognostic value of the pre-operative GRADE score for 5-year survival among older patients undergoing major digestive or non-breast gynecological cancer surgery. 

## 2. Materials and Methods

### 2.1. Study Design and Population

Patients were recruited from the Physical Frailty in Elderly Cancer patients (PF-EC) survey, a prospective observational two-centre cohort study that started in November 2013 [15]. All consecutive in- and outpatients aged 65 and over, referred by a surgeon specialized in cancer for a geriatric oncology assessment, were prospectively included in a registry if a diagnosis of cancer was established, and if frailty was felt, before a cancer treatment decision was made. 

For the present study, we included in the analysis all consecutive patients before major cancer surgery (digestive tract and non-breast gynaecological tract) up to 31 August 2019. All patients underwent a cancer surgery in intention of curative treatment. 

The inclusion date was the date of the first geriatric oncology visit. Informed consent was obtained from the patients before inclusion. The study was approved by the local ethics committee (CLEA-2015-019, Avicenne Hospital, Bobigny, France).

### 2.2. Pre-Operative Assessment

Demographic data (age, gender), in/outpatient status, Eastern Cooperative Oncology Group Performance Status score (ECOG-PS) and American Society Anesthesiology (ASA) scale score were obtained at the first geriatric oncology visit as part of the Geriatric Assessment (GA). Tumour characteristics (site and extension) were also recorded. Tumour extension was classified as local, locally advanced or metastatic.

Vulnerability was assessed at the first geriatric oncology visit and before the GA using the G8 index [16]. The total G8 index score ranges from 0 to 17, and a score ≤ 14 is considered to indicate impairment.

The Geriatric Assessment (GA) was performed at the first geriatric oncology visit and included the following six domains: comorbidities assessed by the cumulate illness rating scale—geriatric (CIRSG) [17] (abnormal if ≥ 14 [median]); polypharmacy (≥5 drugs a day) [18]; dependency (activities of daily living scale (ADL) ≤ 5/6, and/or a instrumental-ADL scale (IADL) ≤ 3/4) [19,20]; malnutrition (body mass index (BMI) < 21 kg/m^2^ and/or an albumin level < 35 g/L) [21]; depressed mood (Mini-Geriatric Depression Scale (Mini-GDS) ≥ 1/4) [22]; and cognitive impairment (Mental State Examination (MMSE) < 24/30) [23]. 

The GRADE score was collected at the first geriatric oncology visit. As described elsewhere, patients were scored as follows: unintentional weight loss in the past year ≥ 5% (no/yes: 0/1) + slow gait speed < 0.8 m/s (no/yes: 0/3) + cancer site (colorectal: 3, digestive non-colorectal: 4, non-breast gynaecological: 3) + cancer extension (local/locally advanced/metastatic 0/3/5) [14]. 

### 2.3. Post-Operative Outcomes

The primary outcome was overall 5-year post-operative mortality. Vital status was determined by calling patients or their families, or from medical records. 

The secondary outcome was severe complications in the 30 days after surgery, defined as a Clavien-Dindo score ≥ 3 [24]. 

### 2.4. Statistical Analysis

Categorical variables were summarized as numbers (percentage), and continuous variables were summarized as means ± standard deviation (SD).

Comparison of patients according to the GRADE score: we divided the GRADE score into 2 groups at a median of 8. Baseline characteristics (demographic data, cancer-related data and GA components) and post-operative complications were compared between these two groups using the Chi2 test. 

For survival analyses, univariate survival curves were plotted according to the Kaplan-Meier method for the GRADE score with the threshold of 8. The discrimination of the models was assessed using Harrell’s C index with 95%CI with the following classification: 0.5–0.59 (poor), 0.6–0.69 (moderate), 0.7–0.79 (good), 0.8–0.89 (very good) and ≥0.9 (excellent). The calibration of the GRADE score was assessed using the Grönnesby and Borgan goodness of fit (a significant *p* value indicates miscalibration). Cox uni- and multivariate proportional hazards regression models were run to identify pre-operative factors associated with 5-year post-operative mortality. The model assumptions were verified including proportional hazards using Schoenfeld residuals. Variables yielding *p* values (log-rank test) under 0.20 in the univariate analysis were considered for inclusion in the multivariate analysis. A stepwise selection at the *p*-level < 0.05 was performed to produce the final multivariate model. The association between pre-operative factors and 5-year mortality was expressed by adjusted hazard ratio (aHR) and 95% CI.

All tests were two-sided, and the threshold for statistical significance was set at *p* < 0.05. The data was analyzed using R statistical software (version 4.0.3, R Foundation for Statistical Computing, Vienna, Austria; http://www.r-project.org, (accessed on 7 November 2021)).

## 3. Results

### 3.1. Patients 

By 31 August 2019, 136 consecutive patients aged 65 and over with digestive or non-breast gynecological cancer who were referred for GA had been selected for major cancer surgery and were prospectively included in this study. 

### 3.2. Baseline Characteristics of Patients

The patients’ age ranged from 65 to 97 years with a median of 80 years. Most patients were men (*n* = 71, 52%), outpatients (*n* = 108, 79%), and most had colorectal cancer (*n* = 99, 73%) and local or locally-advanced cancer (*n* = 107, 79%). The remaining 29 patients were reclassified as metastatic after the surgery procedure. The G8-index was ≤14 for 112/136 (84%) of the patients. According to the tools and thresholds used, impairment in the domains explored by the GA varied from 13% (BMI < 21 kg/m^2^) to 65% (polypharmacy) (Table 1).

Compared to patients with an ECOG-PS ≤ 2 (*n* = 98), patients with an ECOG-PS > 2 (*n* = 38) were significantly (*p* < 0.05) older (mean age of 83.0 years), and they had a higher proportion of ASA scale > 2 (*n* = 29/38, 76%), G8-index ≤ 14 (*n* = 38/38, 100%), CIRSG total ≥ 14 (*n* = 27/38, 71%), ADL ≤ 5/6 (*n* = 29/38, 76%), IADL ≤ 3/4 (*n* = 36/38, 95%), BMI < 21 kg/m^2^ (*n* = 9/38, 24%), and MMSE < 24/30 (*n* = 17/21, 81%).

The GRADE scores ranged from 3 to 13. Patients were classified into two groups defined by the median GRADE score as follows: low-risk (GRADE ≤ 8) and high risk (GRADE > 8). 

In univariate analysis, older age, female gender, cancer extension, ASA scale > 2, ECOG-PS > 2, G8-index ≤ 14, ADL ≤ 5/6, IADL ≤ 3/4, and MMSE < 24/30 were significantly associated with a GRADE score > 8.

### 3.3. 30-Day Post-Operative Complications

The 30-day post-operative complication rate (Clavien-Dindo ≥ 1) was 67% (*n* = 91/136), of which 55% were classified as severe (Clavien-Dindo ≥ 3a). 

Figure 1 shows the characteristics of 30-day post-operative complications according to the Clavien-Dindo scale. Among severe complications (*n* = 91), multiorgan failure was the most frequent and concerned 9 (10%) patients. Six (7%) patients died. Among non-severe complications, sepsis, ileus and confusion were the most frequent, and concerned 9%, 7% and 5.5% of the patients, respectively. A GRADE score >8 was significantly associated with severe post-operative complications (OR = 2.16 [1.06–4.41], *p* = 0.03) (Table 1).

### 3.4. Preoperative Factors Associated with 5-Year Post-Operative Mortality

The median follow-up was 24.2 months (7.0–39.5) [min-max: 0.16–71.5]. On 1 December 2019, the 5-year post-operative mortality rate was 34.5% (*n* = 47/136). Median OS was 65.5 months (38.0-NA). 

In univariate analysis, age as a continuous variable, GRADE score > 8, ASA score > 2, ECOG-PS score > 2, CIRSG total score ≥ 14, ADL score ≤ 5/6, IADL score ≤ 3/4, BMI < 21 kg/m^2^, Mini-GDS score ≥ 1/4, and MMSE score < 24/30 were significantly associated with 5-year mortality (Table 2). Figure 2 shows the Kaplan-Meier survival curves according to the pre-operative GRADE score (≤8: low-risk; >8: high-risk). Median survival was reached at 34 months [95% CI: 19.2–50.1] among patients with GRADE score > 8. It was not reached for patients with GRADE score ≤ 8. At one year, the survival rate was significantly different for patients with a GRADE score ≤ 8 (92%) compared to patients with a GRADE score > 8 (70%) (*p* < 0.0001). At 5 years, among patients with a GRADE score ≤ 8 the survival rate was 71%, compared to 26% for patients with a GRADE score > 8 (*p* < 0.0001) (Table 3). 

In multivariate analysis, factors associated with 5-year post-operative mortality were GRADE score > 8, IADL score ≤ 3/4 and BMI < 21 kg/m^2^. There was no significant interaction across multivariate predictors (Table 2). There was not either significant interaction between the G8-index and the GRADE, IADL or BMI. The discrimination of the GRADE score was good (Harrell’s C index = 0.76 [0.64–0.88]). There was no miscalibration (*p* = 0.48). 

### 3.5. Improvement of the GRADE Score: The GRADE-Surgery Score

In order to improve discrimination, we added the IADL measure to the GRADE scoring system using the Schneeweiss’s beta-coefficient point-based method, which weights by 1 unit more with each 0.3 increase in the beta-coefficient (i.e., GRADE score > 8 no/yes = 0/3; IADL score ≤ 3/4 no/yes = 0/4) [25]. 

The GRADE-IADL score (or the GRADE-surgery score) ranges from 0 to 7 with a median value of 4. The two scoring systems are shown in Table A1. A GRADE-surgery score > 4 remained significantly associated with 5-year mortality after adjustment for BMI (aHR = 4.33 [2.34–8.00], *p* < 0.0001) with very good discrimination (Harrell’s C index = 0.81 [0.71–0.91]), and it evidenced no significant difference from the original level of discrimination of the GRADE score (*p* = 0.12). Again, there was no miscalibration (*p* = 0.30).

## 4. Discussion

A pre-operative GRADE score > 8 was strongly and significantly associated with 5-year mortality among 136 consecutive older patients undergoing major cancer surgery (colorectal, digestive non-colorectal and non-breast gynecological) after adjustment for IADL-dependency and malnutrition (BMI < 21 kg/m^2^). 

The GRADE score is easy to perform in daily surgical practice with only four parameters, cancer site and extension, weight loss and gait speed, and only adds two minutes to a normal consultation [14]. This is the main strength of this score, since it does not require a geriatric assessment and thus it could be of great help to surgeons in the decision whether or not to perform cancer surgery among older adults, especially when a GA is not available. Indeed, in daily practice, data shows that for older patients with cancer, before surgical decision the Geriatric Assessment is performed in less than 10% of cases, mainly due to time and resource limitations [26]. 

Another strength of the GRADE score is that it could capture heterogeneity in the course of aging. Indeed, our study results show that in the high-risk group (GRADE > 8), patients were older, more dependent, had higher ASA and ECOG-PS scores, a more frequently abnormal G8-index, and greater cognitive impairment. Over the post-operative period ranging from 6 months to 7 years, all of these pre-operative factors are known to be associated with poorer survival among older patients undergoing major cancer surgery [10,27,28,29,30,31]. However, the GRADE score remains independently associated with 5-year mortality after adjustment for IADL-dependency and malnutrition (BMI < 21 kg/m^2^). 

One limitation of our score could be its poorer discrimination compared to the similar five-variable PREOP risk score recently published [30]. Although this score could be more discriminant with a C-index = 0.78, its appropriation by surgeons in daily practice could be more difficult since it uses the Nutritional Risk Screening tool which takes into account weight loss, BMI, overall condition and food intake. Its discriminant power was similar to that of the GRADE score (HR = 2.7 [1.7–4.3]). While maintaining the simplicity of use of the GRADE score, by adding the very simple four-item IADL scoring to it (producing the GRADE-surgery score) we were able to improve its predictive value (C-index = 0.81). The GRADE-surgery score remains easy to perform in daily practice, but an external validation is still required. Table A2 compares the existing pre-operative scoring systems among older adults referred for a cancer-surgery [30,32,33]. To date, the GRADE score appears to be the simplest pre-operative tool. However, further studies are required to compare pre-operative scoring systems and to decipher (i) their predictive performances on post-operative complications and overall survival; and (ii) their appropriation by surgeons in daily practice. 

Other limitations deserve to be discussed: first, due to the study design, too much peri-operative data (type, duration and complexity of surgery procedure) was missing to take it into account in multivariate analyses; secondly, the limited sample size and the prevalent colorectal cancer type could immediately limit the generalization of the GRADE score; and thirdly, despite the interest that there would have been in conducting a time-varying Cox model analysis to assess the potential influence on overall survival of the subsequent treatments or complications over a 5-year time frame, we lacked the detailed information required for conducting such an analysis. 

On the basis of our study results, we suggest that older patients with cancer and with a GRADE score ≤8 could be treated according to the standard guidelines, including cancer surgery. Conversely, for those with a GRADE score >8, with their 2-fold increased risk of severe post-operative complications and 2.5-fold increased risk of 5-year post-operative mortality, nutritional and functional rehabilitation could be proposed before considering cancer surgery in order to improve the prognosis. Indeed, in a recent meta-analysis including 3962 participants aged from 55 to 81 years, the authors found that multimodal pre-habilitation was significantly effective on overall post-operative complications after abdominal cancer surgery [34]. 

## 5. Conclusions

The preoperative GRADE score was strongly and significantly associated with 5-year post-operative mortality among older patients undergoing cancer surgery. 

We suggest that surgeons could use this score at the first surgical consultation to propose optimal treatment, especially when a geriatric assessment is not available. 

## Figures and Tables

**Figure 1 cancers-14-00117-f001:**
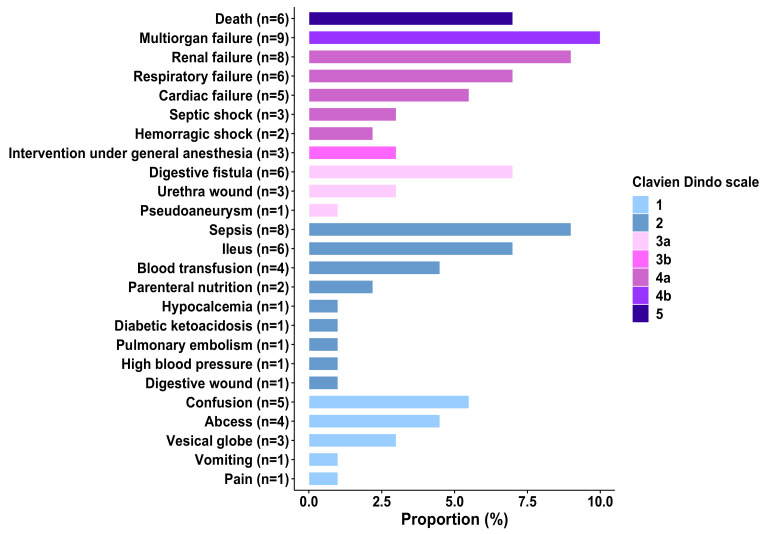
Bar plot of the 30-day post-operative complications among 136 older patients with major cancer surgery.

**Figure 2 cancers-14-00117-f002:**
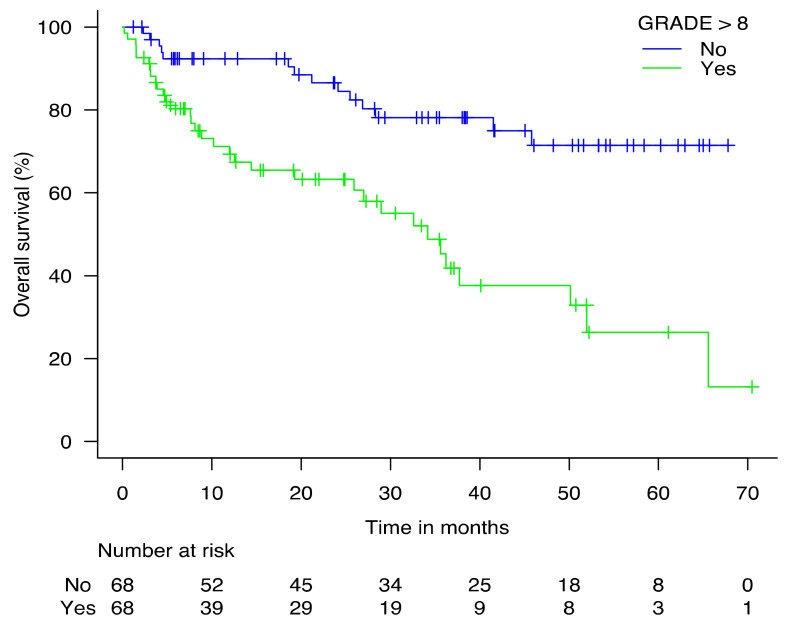
5-year Kaplan-Meier survival according to the GRADE score for 136 older patients with major cancer surgery.

**Table 1 cancers-14-00117-t001:** Baseline characteristics and comparison of 136 older patients with major cancer surgery according to the GRADE score.

Variables	Whole Cohort	GRADE ≤ 8	GRADE > 8	*p* *
		Low-Risk	High-Risk	
	*n* = 136 (%)	*n* = 68 (%)	*n* = 68 (%)	
Age, y				
Mean ± SD	80 ± 7	78 ± 7	82 ± 7	**0.0008**
65–74	32 (23.5)	20 (29)	12 (18)	0.09
75–84	66 (48.5)	34 (50)	32 (47)	
≥85	38 (28)	14 (21)	24 (35)	
Gender (male)	71 (52)	42 (62)	29 (43)	**0.02**
Outpatient (yes)	108 (79)	60 (88)	48 (70)	0.25
Cancer site:				0.25
Colorectal	99 (73)	47 (69)	52 (76)	
Others †	37 (27)	21 (31)	16 (24)	
Local and locally-advanced cancer (yes)	107 (79)	62 (91)	45 (66)	**<0.0001**
ASA scale > 2	60 (44)	21 (31)	39 (57)	**0.001**
ECOG > 2	38 (28)	9 (13)	29 (43)	**0.0001**
G8-index £ 14/17 (*n* = 133)	112 (84)	50 (73)	62 (91)	**0.007**
Comorbidities:				
CIRSG total ≥ 14	67 (49)	29 (43)	38 (56)	0.12
Polypharmacy (yes)	88 (65)	42 (62)	46 (68)	0.47
Dependency				
ADL £ 5/6	45 (33)	12 (18)	33 (48)	**0.0001**
IADL £ 3/4	76 (56)	26 (38)	50 (73)	**<0.0001**
Malnutrition				
BMI < 21 kg/m^2^ (*n* = 134)	18 (13)	8 (12)	10 (15)	0.56
Depressed mood				
Mini-GDS ≥ 1/4	51 (37.5)	22 (32)	29 (43)	0.21
Cognition (*n* = 91)				
MMSE < 24/30	41 (45)	19 (28)	22 (32)	**0.03**
30-day post-operative complications				
Clavien-Dindo ≥ 1	91 (67)	41 (60)	50 (73)	0.1
Clavien-Dindo ≥ 3a (severe)	50 (37)	19 (28)	31 (46)	**0.03**

* χ^2^ test or Fisher’s exact test for categorical variables as appropriate; Bold = significant *p* value (<0.05). †: gastric (*n* = 17); pancreas (*n* = 8); oesophagus (*n* = 3); bile-duct (*n* = 2); gastrointestinal and stromal tumours (*n* = 2); anus (*n* = 1); ovarian (*n* = 3); uterus (*n* = 1). ASA: American Society Anesthesiology; ECOG-PS: Eastern Cooperative Oncology Group Performance Status; CIRSG: Cumulative Illness Rating Scale Geriatric; ADL: Activities of Daily Living; IADL: Instrumental-ADL; BMI: Body Mass Index; Mini-GDS: Mini-Geriatric Depression Scale; MMSE: Mini Mental State Examination.

**Table 2 cancers-14-00117-t002:** Preoperative factors associated with 5-year mortality among 136 older patients with major cancer surgery.

Variables	Univariate Analysis			Multivariate Analysis		
	HR	(95% CI)	*p* *	aHR	(95% CI)	*p* *
Age (per 1 SD of more)	1.05	1.01–1.10	**0.02**	-		
Gender (male)	0.61	0.34–1.09	0.09	-		
Outpatients (yes)	0.55	0.29–1.06	0.07	-		
GRADE score			**0.0001**			
≤8 (low risk)	1 (reference)	–		1 (reference)	-	**0.005**
>8 (high risk)	3.47	1.85–6.54		2.64	(1.34–5.21)	
ASA scale > 2	3.43	1.87–6.30	**<0.0001**	-		
ECOG-PS > 2	3.42	1.88–6.23	**<0.0001**	-		
G8-index £ 14/17 (*n* = 133)	3.09	0.96–10.0	0.05	-		
Comorbidities:						
CIRSG total ≥ 14	1.95	1.09–3.51	**0.02**	-		
Polypharmacy (yes)	1.63	0.86–3.10	0.13	-		
Dependency						
ADL £ 5/6	2.22	1.24–3.98	**0.007**		-	
IADL £ 3/4	4.32	2.13–8.73	**<0.0001**	2.95	(1.40–6.23)	**0.004**
Malnutrition						
BMI < 21 kg/m^2^ (*n* = 134)	2.66	1.35–5.25	**0.004**	2.97	(1.49–5.93)	**0.002**
Depressed mood						
Mini-GDS ≥ 1/4	1.88	1.06–3.33	**0.03**	-		
Cognition (*n* = 91)						
MMSE < 24/30	2.61	1.18–5.73	**0.01**	-		

* *p* value for log-rank test; Bold = significant *p* value (<0.05); ASA: American Society Anesthesiology; ECOG-PS: Eastern Cooperative Oncology Group Performance Status; CIRSG: Cumulative Illness Rating Scale Geriatric; ADL: Activity of Daily Living; IADL: Instrumental-ADL; BMI: Body Mass Index; Mini-GDS: Mini-Geriatric Depression Scale; MMSE: Mini Mental State Examination; NA: Non-Available.

**Table 3 cancers-14-00117-t003:** 5-year risk of death according to the pre-operative GRADE score among older patients undergoing major cancer surgery.

		Risk of Death				
GRADE	Median Survival (Months)	12 m	24 m	36 m	50 m	62 m
≤8 (low risk)	NR	8%	13%	22%	29%	29%
>8 (high risk)	34.2 (19.2–50.1)	31%	37%	55%	62%	74%

NR: not reached.

## Data Availability

The data presented in this study are available on request from the corresponding author.

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
