# Peer review of "The Pre-Operative GRADE Score Is Associated with 5-Year Survival among Older Patients with Cancer Undergoing Surgery"

_cancers, 2021, doi:10.3390/cancers14010117_

Round 1
Reviewer 1 Report
The pre-operative GRADE score is associated with 5-year survival among older patients with cancer undergoing surgery
The authors prospectively evaluated the prognostic value of an easily applicable score, which may predict whether older patients with (mainly colorectal) cancer will benefit of major surgery, or not. Such solutions are really needed and appreciated among health-care professionals taking care of older patients with cancer and the authors deserve huge congratulation. This paper is well written, has appropriate structure, and timely references.
I have some comments and suggestions, please see them below:
Simple summary
“We were able to identify two groups at increasing risk at the threshold of 8 and with a good discrimination.” – Patients having a score over eight were at increasing risk. Two groups were identified by the threshold, but only one of them was at increased risk.
Abstract
IADL: Instrumental Activities of Daily Living – this is the widely accepted term for the abbreviation.
You use postoperative among key words, but post-operative in the main text. Please use one of the forms consistently.
Introduction
Ageing and the increasing number of older adults are global phenomena, rather than only seen in the Western countries.
“We hypothesized that the GRADE score could help decide whether or not to undertake radical treatment, including cancer surgery, for in older patients.” – Other treatment modalities were not investigated, only surgery. Please consider the suggested changes.
M&M
“All consecutive in- and outpatients aged 65 and over, referred by a surgeon specialized in cancer for a geriatric oncology assessment, were prospectively included in a registry if a diagnosis of cancer was established, and if frailty was suspected, before a cancer treatment decision was made.” – in the original PE-FC cohort, 959 pts were included and 139 were considered being candidate for major surgery. If frailty was not suspected, the patient was not referred for GA, but the presence of frailty/vulnerability is often not obvious without applying geriatric screening. How was frailty suspected? Was any validated geriatric screening tool like G8 or VES-13 used, or was it the surgeons gut feeling? It might be considered as a potential selection bias and needs to be explained. (I see G8 is mentioned later in the paper, however, it is unclear whether it was used before or after GA was performed.)
Instead of poly-medication you may consider the use polypharmacy. Actually, you use the term polypharmacy in the results section.
ADL: Activities of Daily Living
If vulnerability was assessed before GA, it should be move before GA in the text as well.
“As previously described, patients were scored as follows: unintentional weight……..” – it was not previously described in the manuscript, rather elsewhere (ref 14.)
Statistical analyses
“…..and post-operative complications were compared between these two groups using the Chi2 test” – here, I rather would consider logistic regression, if you want to explicitly define a dependent variable and make predictions. Would you consider it?
“The model assumptions were verified” – was it done by Schoenfeld residuals? Please add the method.
Results
“Among severe complications (n=91), multiorgan failure was the most frequent and concerned 9/91 patients (10%). 6/91 patients died (7%).” - I rather would formulate this sentence this way: Among severe complications (n=91), multiorgan failure was the most frequent and concerned 9 (10%) patients; six (7%) patients died.
In the next sentence, I would set a comma before “respectively”. The use of Oxford commas can be considered.
Of the 136 patients, 107 had localized or locally advanced cancer. Does it mean that 29 patients had metastatic disease? It is important because patients with metastatic disease usually do not benefit of major surgery and do not get it offered either. The only exception is ovarian cancer. Please provide more details here.
Patients with ECOG PS >2 (3-5) are usually not considered being candidates for active oncological care/major surgery, especially with metastatic disease. Would you please provide more details about the characteristics of the patients presented with ECOG 3-4 and selected for major surgery?
“Overall, median survival was reached at 65.5 months (38.0-NA).” – I would rather write simpler: Median OS was 65.5 months (38-NA).
It should always be a space like > 8 and ≤ 8, or not, like >8 and ≤8. Please use these consistently through the manuscript.
Author Response
Reviewer 1.
The authors prospectively evaluated the prognostic value of an easily applicable score, which may predict whether older patients with (mainly colorectal) cancer will benefit of major surgery, or not. Such solutions are really needed and appreciated among health-care professionals taking care of older patients with cancer and the authors deserve huge congratulation. This paper is well written, has appropriate structure, and timely references.
We would like to thank Reviewer 1 for highlighting the overall quality of our manuscript.
I have some comments and suggestions, please see them below:
Simple summary
“We were able to identify two groups at increasing risk at the threshold of 8 and with a good discrimination.” – Patients having a score over eight were at increasing risk. Two groups were identified by the threshold, but only one of them was at increased risk.
To follow this remark by Reviewer 1, we have now deleted this sentence and we have modified the simple summary.
Abstract
IADL: Instrumental Activities of Daily Living – this is the widely accepted term for the abbreviation.
To follow this remark by Reviewer 1, we have now modified the sentence: Line 45 “Instrumental Activities of Daily Living”.
You use postoperative among key words, but post-operative in the main text. Please use one of the forms consistently.
In agreement with this remark, we have now used consistently the word “post-operative” across the manuscript.
Introduction
Ageing and the increasing number of older adults are global phenomena, rather than only seen in the Western countries.
In agreement with Reviewer 1, we have now modified the first sentence of introduction as follows: Line 53 “With the ageing of populations worldwide…”
“We hypothesized that the GRADE score could help decide whether or not to undertake radical treatment, including cancer surgery, for in older patients.” – Other treatment modalities were not investigated, only surgery. Please consider the suggested changes.
In agreement with Reviewer 1, we have now modified this sentence as follows: Line 82 “We hypothesized that the GRADE score could help decide whether or not to undertake radical cancer surgery in older patients”.
M&M
“All consecutive in- and outpatients aged 65 and over, referred by a surgeon specialized in cancer for a geriatric oncology assessment, were prospectively included in a registry if a diagnosis of cancer was established, and if frailty was suspected, before a cancer treatment decision was made.” – in the original PE-FC cohort, 959 pts were included and 139 were considered being candidate for major surgery. If frailty was not suspected, the patient was not referred for GA, but the presence of frailty/vulnerability is often not obvious without applying geriatric screening. How was frailty suspected? Was any validated geriatric screening tool like G8 or VES-13 used, or was it the surgeons gut feeling? It might be considered as a potential selection bias and needs to be explained. (I see G8 is mentioned later in the paper, however, it is unclear whether it was used before or after GA was performed.)
The study design allowed surgeons to refer older patients for geriatric assessment (GA) based on their gut feeling of frailty. Then, the G8-index was performed during the first patient’s geriatric consultation before the GA.
We have now clarified this point in METHODS section: Line 92“All consecutive in- and outpatients aged 65 and over, referred by a surgeon specialized in cancer for a geriatric oncology assessment, were prospectively included in a registry if a diagnosis of cancer was established, and when a frailty was felt, before a cancer treatment decision was made.
We have also modified the sentence about the G8-index as follows: “Vulnerability was assessed at the first geriatric oncology visit and before the GAusing the G8 index [23].
Instead of poly-medication you may consider the use polypharmacy. Actually, you use the term polypharmacy in the results section.
In agreement with Reviewer 1, we have now corrected the term across the manuscript as follows: “polypharmacy”
ADL: Activities of Daily Living
To follow this remark by Reviewer 1, we have now modified the sentence: Line 112 “Activities of Daily Living”.
If vulnerability was assessed before GA, it should be move before GA in the text as well.
We have now moved (Line 106) the paragraph before the GA chapter.
“As previously described, patients were scored as follows: unintentional weight……..” – it was not previously described in the manuscript, rather elsewhere (ref 14.)
We have now modified this sentence as follows: Line 116“As described elsewhere…”
Statistical analyses
“…..and post-operative complications were compared between these two groups using the Chi2 test” – here, I rather would consider logistic regression, if you want to explicitly define a dependent variable and make predictions. Would you consider it?
We thank Reviewer 1 for this remark, but here we did not want to provide a predictive model. The comparison between low- and high-risk groups is able to understand a phenotype associated with each of them. Conversely, a logistic regression would suggest a risk ratio of belonging to one or the other category. It was not our purpose.
“The model assumptions were verified” – was it done by Schoenfeld residuals? Please add the method.
We have now added this point as follows: Line 141 “The model assumptions were verified including proportional hazards using Schoenfeld residuals”.
Results
“Among severe complications (n=91), multiorgan failure was the most frequent and concerned 9/91 patients (10%). 6/91 patients died (7%).” - I rather would formulate this sentence this way: Among severe complications (n=91), multiorgan failure was the most frequent and concerned 9 (10%) patients; six (7%) patients died.
To follow this remark of Reviewer 1, we have now included these corrections in RESULTS section (Line 177).
In the next sentence, I would set a comma before “respectively”. The use of Oxford commas can be considered.
We have now added a comma before “respectively” (Line 179).
Of the 136 patients, 107 had localized or locally advanced cancer. Does it mean that 29 patients had metastatic disease? It is important because patients with metastatic disease usually do not benefit of major surgery and do not get it offered either. The only exception is ovarian cancer. Please provide more details here.
In our study, the 29 patients were classified as metastatic after the surgery procedure. We have now emphasized this point as follows: Line 157 “and most had colorectal cancer (n=99, 73%) and local or locally-advanced cancer (n=107, 79%). The remaining 29 patients were classified as metastatic after the surgery procedure. The G8-index… ».
Patients with ECOG PS >2 (3-5) are usually not considered being candidates for active oncological care/major surgery, especially with metastatic disease. Would you please provide more details about the characteristics of the patients presented with ECOG 3-4 and selected for major surgery?
To follow the advice of Reviewer 1, we added more details about patients with an ECOG > 2 as follows: Line 161 “Compared to patients with an ECOG-PS ≤ 2 (n=98), patients with an ECOG-PS > 2 (n=38) were significantly (P < 0.05) older (mean age of 83.0 years), and they had a higher proportion of ASA scale > 2 (n=29/38, 76%), G8-index ≤ 14 (n=38/38, 100%), CIRSG total ≥ 14 (n=27/38, 71%), ADL ≤ 5/6 (n=29/38, 76%), IADL ≤ 3/4 (n=36/38, 95%), BMI < 21 kg/m2 (n=9/38, 24%), and MMSE < 24/30 (n=17/21, 81%)”.
“Overall, median survival was reached at 65.5 months (38.0-NA).” – I would rather write simpler: Median OS was 65.5 months (38-NA).
We have now corrected this sentence as suggested by Reviewer 1 (Line 192)
It should always be a space like > 8 and ≤ 8, or not, like >8 and ≤8. Please use these consistently through the manuscript.
We have now corrected the spaces consistently through the manuscript.

Reviewer 2 Report
The authors investigated the role of the GRADE score and the GRADE score plus IADL to predict 5 year mortality in a sample of older adults with cancer mainly affected by colorectal cancer. The GA assessment was also perfomed at baseline.
Form a methodological point of view there are some major criticisms.
-Both the GRADE and GRADE IADL score included the cancer site and extension ad clinical variables. These variables should instead adjusted for in the multivariable analysis rather than included subitems . It is common knowledge that the extension and type of cancer predict cancer related mortality. With regard to this, it is unclear wether the authors measured overall mortality or cancer related mortality. Moreover, the multivariable analysis should be also adjusted for frailty stratification at baseline.
-The curative intent or palliative intent of surgery should be specified along with peri-operative information ( type and duration of surgery, procedure complexity according to objective assessment such as WRVU criteria)
- The limited sample size and the prevalent colorectal cancer type limit the generalization of the findings
-5 years mortality seems a fair inappropriate outcome for a surgical calculator. A competing risk model controlling for the huge series of intervening clinical variables in such a long observational period should be performed.
-The analysis of clinical peri-operative variables should also take into account , including short term geriatric complications ( falls, delirium...), post discharge needs, including functional status at hospital discharge
-it is unclear the added value of this surgical calculator compared to the existing ones . The authors should perform the analysis comparing for a gold standard to support the added value of GRADE plus
- The study design is not meant to replace GA assessment although the authors seems to discuss this issue at least partially
-The discussion is poor and a full critical discussion on all the existing surgical tools should be implemented. For instance the vESPA toll among others is not mentioned. The conclusions are poor and poorly support the results
Author Response
Reviewer 2.
The authors investigated the role of the GRADE score and the GRADE score plus IADL to predict 5 year mortality in a sample of older adults with cancer mainly affected by colorectal cancer. The GA assessment was also performed at baseline.
Form a methodological point of view there are some major criticisms.
-Both the GRADE and GRADE IADL score included the cancer site and extension as clinical variables. These variables should instead adjusted for in the multivariable analysis rather than included subitems. It is common knowledge that the extension and type of cancer predict cancer related mortality.
As scoring systems, the GRADE and GRADE IADL are able to capture multiple variables which are known to be associated with survival in cancer patients. Thus, cancer site and cancer extension were indeed considered during the multivariate analysis by using the GRADE score.
With regard to this, it is unclear wether the authors measured overall mortality or cancer related mortality.
This is the overall mortality which was assessed as the first outcome. We have now added this information in METHODS section as follows: Line 122 “The primary outcome was overall 5-year post-operative mortality”.
Moreover, the multivariable analysis should be also adjusted for frailty stratification at baseline.
We disagree with Reviewer 2, since frailty was assessed at baseline using the G8-index. The G8-index has been included in univariate and multivariate analyses and was not found to be a multivariate predictor of overall survival. In addition, we checked the absence of a significant interaction between G8/GRADE, G8/IADL and G8/BMI. For these methodological reasons, the frailty stratification at baseline was not considered.
We have now added these data in RESULTS section as follows: Line 217“There was not either significant interaction between the G8-index and the GRADE, IADL or BMI”.
-The curative intent or palliative intent of surgery should be specified along with peri-operative information ( type and duration of surgery, procedure complexity according to objective assessment such as WRVU criteria)
“All patients underwent a cancer surgery in intention of curative treatment”. We have now added this information in METHODS section (Line 96).
Due to the study design, peri-operative information resulted in too much missing data. We have now added it as a limit of our study in DISCUSSION section as follows: Line 267“Other limitations deserve to be discussed: first, due to the study design, too much peri-operative data (type, duration and complexity of surgery procedure) was missing to take it into account in multivariate analyses…”
- The limited sample size and the prevalent colorectal cancer type limit the generalization of the findings
Reviewer 2 is right. An external validation of the GRADE is required to make it generalizable. We have now added it as a limit of our study in DISCUSSION section as follows: Line 269 “…the limited sample size and the prevalent colorectal cancer type could immediately limit the generalization of the GRADE score…”
-5 years mortality seems a fair inappropriate outcome for a surgical calculator. A competing risk model controlling for the huge series of intervening clinical variables in such a long observational period should be performed.
We agree with the Reviewer 2 that there would be great interest in assessing the potential influence on overall survival of the subsequent treatments or complications occurring during such a long follow-up. This analysis would have been complementary to our main objective of a prognostic analysis based on baseline information only, but we unfortunately lacked the detailed information required for conducting such a time-varying Cox model on 5-year OS in our database.
To account for the reviewer's comment, we added the following sentence to the DISCUSSION section relating to the limits of the study: Line 271 "Despite the interest that there would have been in conducting a time-varying Cox model analysis to assess the potential influence on overall survival of the subsequent treatments or complications over a 5-year time frame, we lacked the detailed information required for conducting such an analysis"
-The analysis of clinical peri-operative variables should also take into account , including short term geriatric complications ( falls, delirium...), post discharge needs, including functional status at hospital discharge
Unfortunately, these data are not available and this is a limitation of this kind of study. We have now added this limit in DISCUSSION section as follows: “Line 267“Other limitations deserve to be discussed: first, due to the study design, too much peri-operative data (type, duration and complexity of surgery procedure) was missing to take it into account in multivariate analyses…”
-it is unclear the added value of this surgical calculator compared to the existing ones . The authors should perform the analysis comparing for a gold standard to support the added value of GRADE plus
We have now added the need to compare existing scores to the GRADE in dedicated further studies in DISCUSSION section as follows: Line 262 “Table A2 compares the existing pre-operative scoring systems among older adults referred for a cancer-surgery [30, 32, 33]. However, further studies are required to compare pre-operative scoring systems and to decipher i) their predictive performances on post-operative complications and overall survival; and ii) their appropriation by surgeons in daily practice”.
- The study design is not meant to replace GA assessment although the authors seems to discuss this issue at least partially
We do not agree with Reviewer 2. We do not support the idea to replace GA in older patients with cancer. On the contrary, when a GA is not available, the GRADE score could easily help clinicians in their therapeutic decision. We have now emphasized this point in DISCUSSION section as follows: Line 239 “This is the main strength of this score, since it does not require a geriatric assessment and thus it could be of great help to surgeons in the decision whether or not to perform cancer surgery among older adults, especially when a GA is not available”.
-The discussion is poor and a full critical discussion on all the existing surgical tools should be implemented. For instance the vESPA toll among others is not mentioned. The conclusions are poor and poorly support the results
We preferentially compared our score to the similar PREOP score. Nevertheless, other scoring systems exist. To our knowledge, three pre-operative scores among older adults referred for a cancer-surgery have been published: the PREOP, the VESPA and the GA-GYN.
To follow the advice of Reviewer 2, we have now added Table A2 as appendix which compares available pre-operative scoring systems for older adults referred for a cancer-surgery. We have added it in DISCUSSION section as follows: Line 262 “Table A2 compares the existing pre-operative scoring systems among older adults referred for a cancer-surgery [30,32,33]. To date, the GRADE score appears to be the simplest pre-operative tool. However, further studies are required to compare pre-operative scoring systems and to decipher i) their predictive performances on post-operative complications and overall survival; and ii) their appropriation by surgeons in daily practice”.
Round 2
Reviewer 2 Report
The authors failed to appropriately address the raised concerns. The article has serious flaws and there are too many missed clinical variables for adjustment and methodological flaws that seriously hamper the generalization of the findings. The discussion and the novelty of the findings are still poorly discussed.
Author Response
We thank Reviewer 2 for his/her comments.
However, we do not understand the response of reviewer 2 since we addressed all concerns. We agree our work has limitations, notably due to missing information of some clinical variables especially in the perioperative and over the 5-year folllow-up time frame. These limitations are explicitly discussed in the appropriate section of the discussion section.
